# Current Advances in Immunotherapy Management of Esophageal Cancer

**DOI:** 10.3390/cancers17050851

**Published:** 2025-03-01

**Authors:** Sagar Pyreddy, Sarah Kim, William Miyamoto, Zohray Talib, Dev A. GnanaDev, Amir A. Rahnemai-Azar

**Affiliations:** 1School of Medicine, California University of Science and Medicine, Colton, CA 92324, USA; sagar.pyreddy@md.cusm.edu (S.P.); sarah.kim@md.cusm.edu (S.K.); william.miyamoto@md.cusm.edu (W.M.); zohray.talib@cusm.edu (Z.T.); 2Department of Surgery, Arrowhead Regional Medical Center, Colton, CA 92324, USA; gnanadevd@cusm.org; 3Division of Surgical Oncology, Department of Surgery, Arrowhead Regional Cancer Center, California University of Science and Medicine, Colton, CA 92324, USA

**Keywords:** esophageal cancer, immunotherapy, PD-1/PD-L1, immune checkpoint inhibitors, emerging therapy

## Abstract

Advanced esophageal cancers have considerable mortality rates worldwide. Although current treatments include a combination of chemotherapy with surgical intervention, recent advancements in immunotherapy offer new hope for treatment. This article provides a comprehensive summary of various immunotherapy drugs and ongoing trials, exploring their efficacy in managing locally advanced and metastatic esophageal cancers. Additionally, the paper discusses potential future biomarkers that would serve as targets for upcoming immunotherapies that may bring forward additional options of treatment. Despite existing challenges, ongoing investigations are yielding promising results that could potentially reshape future approaches to treatment.

## 1. Introduction

Esophageal cancer (EC) is the 7th most common and 6th most deadly cancer in the world [1]. EC typically occurs as squamous cell carcinoma (ESCC), arising from stratified squamous epithelium in the upper two-thirds part of the esophagus, and adenocarcinoma (EAC), the result of metaplasia, wherein columnar glandular cells replace the typical esophageal squamous epithelium in the lower third [2]. Worldwide, SCC is the predominant type of EC. However, the rates of esophageal AC have risen in high-income countries due to higher incidence of gastroesophageal reflux disease (GERD) and an increase in individuals with excess body weight, and it is now the predominant histologic type of EC in the United States.

Current approaches in the management of EC include endoscopic and surgical resection, radiation therapy (RT), chemotherapy (CT), and targeted therapies with limited application. In ESCC and EAC patients with limited early-stage disease (PTis or pT1a) without metastatic disease, endoscopic therapy is preferred [3]. For cases presenting as locally advanced resectable disease (pT1b, cT1b-cT2, and N0 low-risk lesions), a combination of neoadjuvant therapy followed by surgical intervention is recommended [3,4,5]. In most cases, however, EC is frequently diagnosed in more advanced stages, which has a poor prognosis with a 5-year survival rate of approximately 20% that sharply decreases to 5% in the presence of metastases [6]. In cases where resection is not feasible or the disease has metastasized, systemic therapy is indicated, with typical treatment consisting of a combination of two chemotherapy agents.

With recent FDA approvals for immunotherapy treatments in esophageal cancers, there has been growing attention towards the use of these agents in the treatment of a variety of cancers [7]. While there are still concerns regarding their side effects, immunotherapeutics have proven across multiple trials to be extremely beneficial with close monitoring. Research findings have shown this class of medications to be particularly effective in the management and treatment of locally advanced/metastatic carcinomas [7]. Here, we review the efficacy of some of the most effective immunotherapy medications along with promising emerging and current therapies in treating locally advanced/metastatic upper esophageal cancer.

## 2. Surgical Interventions

Surgical intervention is the only potentially curative option for patients with esophageal cancer; however, its efficacy decreases dramatically with the staging of the cancer. Generally, patients with stage 0 or I are treated with surgery only. Stage II and III are treated with surgery and neoadjuvant therapy. Stage IV patients are generally poor surgery candidates and are mostly treated with chemotherapy, radiation, and immunotherapy. Surgical intervention is proven to be more effective in clearly demarcated cancers [8]. Patients are generally evaluated based on their age, nutritional status, functional status, history of cancer, and prior effects of chemo- and radiotherapy. As surgery is shown to increase survival rates for qualifying patients with esophageal cancer, it is generally performed for those with stage 0-III tumors.

## 3. Immunotherapies

### 3.1. PD-1/PD-L1 Inhibitors

PD-1 is a cell surface inhibitory receptor, which plays a significant physiological role in the maintenance of peripheral immune tolerance [9]. PD-1/PD-Ls is a receptor-ligand system that is part of the adaptive immune response expressed by T-cells during cancer proliferation. Binding of PD-L1 on the surface of cancer cells to the PD-1 receptor on T-cells prevents the initiation of the programmed death pathway and subsequent apoptosis [9]. Immune evasion is a central tactic used by aggressive cancers to increase survival probability and proliferation. Crosstalk via the PD-1/PD-L1 receptor binding complex is an essential part of T-cell activation and subsequent immune response (Figure 1).

#### 3.1.1. Pembrolizumab

Pembrolizumab is a highly selective IgG4-κ humanized monoclonal antibody that prevents PD-1 binding with PD-L1/PD-L2, used in the treatment of a variety of cancers [11]. First approved for use in locally advanced/metastatic esophageal cancer by the FDA in 2019, pembrolizumab is administered in patients whose tumors express PD-L1 CPS (combined positive score) > 10 [12,13]. It is one of the most widely used immunotherapeutics and has demonstrated immense survival benefit and tolerability in the treatment of various cancers.

In 2021, the KEYNOTE-590 trial was the first randomized phase III trial to establish pembrolizumab plus chemotherapy as first-line treatment for esophageal cancer by showcasing significantly increased overall survival (OS) and progression-free survival (PFS). Previous studies such as KEYNOTE-ß028 and KEYNOTE-180/181 laid the groundwork for this by establishing pembrolizumab plus chemotherapy as third- and second-line therapy for advanced esophageal cancer, respectively. A total of 749 patients were enrolled, and pembrolizumab plus chemotherapy was shown to be superior to placebo plus chemotherapy for overall survival in all patients with PD-L1 CPS of 10 or more (12.4 months vs. 9.8 months; *p* < 0.0001) [14,15]. Treatment-related adverse events of grade 3 or higher occurred in 266 (72%) patients in the pembrolizumab plus chemotherapy group versus 250 (68%) in the placebo plus chemotherapy group. This data cemented pembrolizumab plus chemotherapy as first-line treatment for esophageal cancer.

Several early studies have suggested that pembrolizumab can also be highly effective in a neoadjuvant setting. The PALACE-1 trial displayed the safety and efficacy of pembrolizumab as neoadjuvant preoperative therapy for resectable ESCC. Among the 18 patients that underwent surgery, a pathological complete response (pCR) of 55.6% and major pathological response (MPR) of 89% were obtained. The PEN-ICE trial showed a combination of neoadjuvant immunotherapy plus chemotherapy for ESCC is associated with a high immune response in the TME (tumor microenvironment). A total of 13 patients were assessed postoperatively, and a major pathological response (MPR) was noted in 9 cases (9/13, 69.2%) [16]. These trials are indicative of its potential to prove very effective in a neoadjuvant setting; however, phase III trials have yet to be done.

#### 3.1.2. Camrelizumab

Camrelizumab is China’s first immunotherapy drug for EC. A monoclonal, human IgG4 antibody and PD-1 inhibitor, the drug is currently approved in China by the National Medical Products Administration (NMPA) for second-line treatment of advanced or metastatic ESCC. Patients who are treated with first-line chemotherapy and have progressed or become intolerable can then be treated with camrelizumab [17].

The approval is based on studies such as the phase III trial ESCORT, in which patients with advanced or metastatic ESCC who had become intolerant to, or progressed on from, first-line chemotherapy (such as taxanes, cisplatin, or fluorouracil) received second-line camrelizumab. A total of 228 patients received camrelizumab treatment, while 220 received chemotherapy. Results showed a more significantly improved OS in the camrelizumab group than the group given second-line chemotherapy in the form of docetaxel or irinotecan (8.3 months vs. 6.2 months; *p* < 0.001) [17]. In another ESCORT trial, patients with untreated metastatic ESCC were given camrelizumab in combination with chemotherapy as a first-line treatment. Of the 596 patients treated, 490 discontinued the study, primarily due to disease progression or treatment-related adverse effects. From the remaining patients, PFS (6.9 months vs. 5.6 months; *p* < 0.001) and median OS (15.3 months vs. 12.0 months; *p* < 0.001) were significantly greater in the camrelizumab plus chemotherapy group compared to the placebo plus chemotherapy group [18]. These results suggest a benefit of using first-line camrelizumab in conjunction with chemotherapy, in addition to its traditional use as a second-line treatment.

Furthermore, a meta-analysis comparing efficacy and safety of PD-1 inhibitors in ten trials in China found that camrelizumab plus chemotherapy often ranked first in terms of OS and PFS as a first-line treatment, as well as in refractory patients with ESCC [19]. As this drug is yet to be FDA-approved, this information and the promising results of the ESCORT trials mentioned above suggest the need for further consideration and studies in the United States for camrelizumab as a treatment for ESCC.

#### 3.1.3. Sintilimab

Sintilimab is a fully human IgG4 monoclonal antibody (mAb) that binds to PD-1, preventing its interaction with PD-L1 and PD-L2. In preclinical settings, it was shown that sintilimab had higher affinity to human PD-1 and slower dissociation rates compared to pembrolizumab and nivolumab. A study done on humanized mouse models further demonstrated that sintilimab exerted a more potent antitumor effect compared to pembrolizumab and nivolumab, accompanied by elevated levels of CD3+ T-cells, CD8+ T-cells, and ratio of CD8+ to Treg tumor-infiltrating lymphocytes [20]. While pembrolizumab and nivolumab are currently FDA approved for regimens in advanced EC, these findings present the potential of sintilimab to be used in future cancer immunotherapy treatments.

In the ORIENT-15 Phase III trial, sintilimab or placebo was given in combination with chemotherapy (cisplatin plus paclitaxel or cisplatin plus 5-fluorouracil) as a first-line treatment for advanced or metastatic ESCC. A total of 659 participants were randomly assigned sintilimab (n = 327) or placebo (n = 332) with combination therapy. Regardless of PD-L1 CPS level, patients given sintilimab presented with improved OS compared to that of patients given the placebo (16.7 vs. 12.5 months; *p* < 0.001). Adverse effects (grade 3–4) were experienced by both groups, with the sintilimab-chemotherapy group being slightly higher than the placebo-chemotherapy group (60% vs. 55%) [21]. One limitation to this trial was the lack of patients enrolled outside of China (n = 19, 7%). Thus, the data is mainly representative of Chinese patients with advanced ESCC.

Sintilimab as a second-line therapy in advanced ESCC was studied in the ORIENT-2 Phase II trial. Patients were randomized to receive sintilimab either as monotherapy (n = 94) or chemotherapy (n = 87, paclitaxel or irinotecan). The median OS in the sintilimab group was 7.2 months, compared to 6.2 months in the chemotherapy group (*p* = 0.032). Treatment-related adverse events were lower in the sintilimab group vs. chemotherapy group (54.3% vs. 90.8%) [22]. Results of this study indicate potential use of sintilimab for advanced ESCC patients who have progressed after first-line therapy. However, further investigation of this study is necessary due to the small sample size and the study’s restriction to the Chinese population.

#### 3.1.4. Toripalimab

Toripalimab is a humanized IgG4 mAb that targets PD-1. The JUPITER-06 Phase III trials was a multicenter, randomized study in China that demonstrated the efficacy of toripalimab plus chemotherapy (paclitaxel plus cisplatin) as a first-line treatment for patients with advanced ESCC. A total of 514 patients were either given toripalimab plus chemo (n = 257) or placebo plus chemo (n = 257). Results of the study revealed significantly higher OS in the toripalimab-chemo group vs. the placebo-chemo group (17 vs. 11 months; *p* = 0.0004), and the 1-year OS rates were 66% and 44%, respectively. Moreover, there was significant improvement in the ORR, DoR, and DCR in the toripalimab group compared to the placebo group. Treatment-emergent adverse events (TEAE) were comparable across both groups, with grade 3 and above TEAEs present in 73% in the toripalimab group and 70% in the placebo group [23]. Similar to the ORIENT-15 trials, the JUPITER-06 trials focused solely on Chinese patients with advanced ESCC, which limits the ability to draw generalized conclusions about this therapy.

Additionally, a network meta-analysis of five trials in China was performed to compare treatment outcomes of different immunotherapies in the first-line setting for advanced or metastatic ESCC. In this analysis, toripalimab in combination with chemotherapy was determined to be ranked first for OS compared to its counterparts, including pembrolizumab, nivolumab, camrelizumab, and sintilimab [24]. Implications of these findings show that toripalimab may be a promising immunotherapy option for patients with advanced EC. Still, further research must be conducted to fully understand the benefits and limitations of this drug. Currently, a phase III trial is being done in China to study the effectiveness of toripalimab as a neoadjuvant treatment in ESCC [25].

#### 3.1.5. Nivolumab

Nivolumab, a human IgG4 monoclonal antibody, is a PD-1 inhibitor used in the treatment of esophageal cancers. Currently, nivolumab is FDA approved as a first-line systemic therapy for advanced EAC, in combination with fluoropyrimidine and platinum-based chemotherapy in HER2-overexpression-negative tumors. It is also approved in combination with fluoropyrimidine and platinum-based chemotherapy or in conjunction with ipilimumab, a CTLA-4 inhibitor, for the treatment of advanced ESCC [26].

The efficacy of nivolumab as a monotherapy for treating advanced ESCC was demonstrated in the ATTRACTION-1 phase II trials, in which 65 patients with metastatic ESCC received a median of 3 cycles of the drug. Upon assessment, ORR was 11 patients or 17% of subjects (95% CI: 10–28%), median OS was 10.78 months (95% CI: 7.4–13.3 months), and median PFS was 1.5 months (95% CI: 1.4–2.8 months). At follow-up, three patients achieved a complete response and survived more than five years [27]. These data suggest early, yet promising, anti-tumor efficacy of nivolumab monotherapy.

Nivolumab has also been shown to be useful as an adjuvant therapy in patients with EAC and ESCC following neoadjuvant chemotherapy and resection. In this double-blind, placebo-controlled, phase III trial, patients with resected stage II or III esophageal or esophageal junction cancer were assigned to receive nivolumab or a matching placebo. The median DFS was significantly greater in the 532 patients in the nivolumab group at 22.4 months (95% CI: 16.6–34.0 months) vs. 11.0 months in the 262 patients in the placebo group (95% CI: 8.3–14.3 months) [28].

As mentioned previously, nivolumab can also be used in combination with chemotherapy and has been shown in multiple studies to be more effective in treating advanced esophageal cancer than chemotherapy alone. In two phase III trials, CheckMate 648 and CheckMate 649, patients with advanced, unresectable EAC and ESCC treated with first-line nivolumab plus chemotherapy demonstrated significant improvements in OS and PFS versus patients who received chemotherapy alone [16,26].

#### 3.1.6. Tislelizumab

Tislelizumab, a humanized IgG4mAb with high affinity for PD-1, was an FDA-approved treatment used in 2024 for unresectable ESCC after prior systemic therapy not involving a PD-1/PD-L1 inhibitor [29]. This decision was based on the RATIONAL 302 study, which demonstrated the potential use of tislelizumab as a monotherapy for patients with advanced ESCC with progression after first-line systemic therapy. The RATIONALE-302 study was a randomized phase III clinical trial that studied the efficacy of tislelizumab vs. a single agent chemotherapy in 512 patients across 11 countries in Asia, Europe, and North America. Analysis showed that the tislelizumab group (n = 256) had a significantly improved OS compared to the chemotherapy group (n = 256), with median OS being 8.6 months (95% CI, 7.5 to 10.4) vs. 6.3 months (95% CI, 5.3 to 7.0), respectively (HR, 0.70; 95% Cl, 0.57–0.85; *p* = 0.0001) [30]. Overall, fewer patients in the tislelizumab group experienced grade 3 or higher TRAE compared to the chemotherapy group (18.8% vs. 55.8%). The most common treatment-related adverse effects (TRAE) with tislelizumab were elevated aspartate aminotransferase (11.4%), anemia (11.0%), and hypothyroidism (10.2%), while the most common in chemotherapy were decreased white blood cell count (40.8%), decreased neutrophil count (39.2%), and anemia (34.6%). This trial was the first study that showed the efficacy of a PD-1 inhibitor in a global population that enrolled patients from both Asia and Europe/North America. Although previous studies of PD-1 inhibitors mainly enrolled Asian patients, RATIONALE-302 noted that survival benefits were present in both Asian (HR 0.72 [95% CI, 0.59–0.90]) and White patients (HR 0.53 [95% CI, 0.32–0.87]) [30].

The RATIONAL-306 trial was another global randomized, double-blind phase 3 study to investigate the efficacy of tislelizumab plus chemotherapy compared with placebo plus chemotherapy in the treatment of advanced ESCC. Median OS in the tislelizumab arm was 17.2 months (95% CI, 15.8–20.1), while OS in the placebo arm was 10.6 months (95% CI, 9.3–12.1) (HR. 0.66 [95% CI, 0.54–0.80]) [31]. Consistent with other trials evaluating other PD-1 inhibitors combined with chemotherapy in ESCC, the safety profile and adverse effects were also comparable between the tislelizumab and placebo groups [31]. Both the RATIONAL-302 and RATIONAL-306 study presented promising results of PD-1 inhibitor antibody therapy in treatment-unresponsive ESCC.

### 3.2. CTLA-4 Inhibitors

As with PD-1, cytotoxic T-lymphocyte-associated-antigen 4 (CTLA-4) is a protein found on the surface of T-lymphocytes that negatively regulates T-cell immune function [32]. Binding of B7 from antigen-presenting cells to CTLA-4 inhibits T-cell activation. While PD-1 primarily functions to suppress T-cells later in the immune response, CTLA-4’s action occurs primarily in lymph nodes earlier in the immune response. Antibody binding to CTLA-4, preventing its association with B7, is the basis for immunotherapy used in the treatment of certain tumors.

#### Ipilimumab

In addition to being used in combination with chemotherapy, nivolumab is also approved for therapy in conjunction with the CTLA-4 inhibitor ipilimumab, another human monoclonal antibody. CheckMate 649 demonstrated a significantly longer median overall survival in patients with advanced EAC who received first-line treatment with nivolumab plus ipilimumab than in the group given only chemotherapy. This includes patients with tumor-cell PD-L1 expression of 1% or greater (13.7 vs. 9.1 months; *p* < 0.0001), as well as the overall population (12.7 vs. 10.7 months *p* < 0.0001) [33]. Interestingly, OS was greater in patients who received nivolumab plus chemotherapy than those given nivolumab plus ipilimumab, but this difference was insignificant in both groups.

The combination of nivolumab and ipilimumab has also been studied as a neoadjuvant therapy. NEONIPIGA, a phase II study, evaluated the efficacy of neoadjuvant nivolumab and ipilimumab in 32 patients with locally advanced, resectable gastric/GEJ junction AC with deficient mismatch repair (dMMR). Patients underwent surgery and were then treated with adjuvant nivolumab. Results showed no unexpected toxicity and a high pCR rate in patients who completed the regimen [34]. Future studies investigating the efficacy of this regimen in patients with EC are needed.

## 4. Programmed T-Cell Therapies

Current therapies are focused on targeting receptor complexes between T-cells and cancer cells; however, a new branch of therapy focused on engineering T-cells to recognize and eliminate cancer has shown promising results in preliminary trials. Chimeric antigen receptor (CAR) T-cells are modified cells that are able to target cells displaying a specific antigen. While CAR T-cell therapy has had incredible clinical responses, it still has to overcome hurdles to become a mainstream therapy, including life-threatening CAR T-cell-associated toxicities, limited efficacy against solid tumors, and resistance in B-cell malignancies, etc. [35,36,37]. However, they have produced respectable results with liquid tumors; therefore, their application to solid tumors is only a matter of time. With new targets (CD22 and CD30) being researched, there is a bright future for this branch of immunotherapy.

## 5. Prognostic Biomarkers

Developing ways to better monitor and understand physiological responses to immunotherapy is important to our ability to monitor and continue care. Recent research has shown that hyper-mutated circulating tumor DNA (ctDNA) may be one such biomarker. In a trial done in 2017, 69 patients who received checkpoint-based immunotherapy were assessed. Of the patients who received checkpoint-based immunotherapy, there was a lower count of ctDNA compared to patients who did not receive checkpoint-based immunotherapy, which is directly associated with a poorer prognosis [38,39,40].

## 6. Future Targets

While many therapeutics currently focus on targeting PD-1/PD-L1 and CLTA-4, emerging research has revealed additional biomarkers that could potentially be utilized as future targets for upcoming cancer immunotherapies.

### 6.1. TIGIT

TIGIT is an immune receptor located on the membrane of some T-cells (CD8+, CD4+, regulatory T-cells, and follicular T helper cells) and natural killer (NK) cells that plays an important role in active and innate immunity. TIGIT binds to two different ligands, CD155 and CD112, which are expressed by tumor cells and antigen-presenting cells. When bound, TIGIT serves as a negative regulator for T-cells, similar to the PD-1/PD-L1 ligand complex [41]. It has been shown that TIGIT plays an important role in the regulation of tumor identification in T-cells both in vivo and in vitro.

Currently, there are multiple trials assessing the viability of immunotherapies that target this receptor domain. Tiragolumab is a human IgG1/kappa monoclonal antibody that has been used in conjunction with atezolizumab for the treatment of non-small cell lung cancer in the CITYSCAPE trial. In this trial, a total of 270 patients were split into two groups of tiragolumab plus atezolizumab and atezolizumab plus placebo. They found median progression-free survival was 5·4 months in the tiragolumab plus atezolizumab group versus 3·6 months in the placebo plus atezolizumab group [41]. It has also been tested in conjunction with atezolizumab for the treatment of metastatic esophageal cancer. The LBA-5 phase Ib trial showed tiragolumab plus atezolizumab therapy was well tolerated, with a suitable safety profile. Furthermore, it showed antitumor activity in patients with metastatic esophageal cancer that had not previously received checkpoint-based immunotherapy [41].

There are other pharmaceuticals that target the TIGIT domain, including ociperlimab, and domvanalimab, etc., which are still in clinical trials, which goes to show that there is significant evidence to show that targeting this domain could prove effective particularly when combined with PD-1 inhibitors in combating locally advanced/metastatic esophageal cancer.

### 6.2. VISTA

V-domain immunoglobulin (Ig) suppressor of T-cell activation (VISTA) is a checkpoint regulator that regulates T-cell proliferation and cytokine production. Recent VISTA sequence analysis and crystal structure investigations have revealed its independent and unique function as compared with B7 family members, such as PD-1 [42]. While other members of the B7 family are found on activated T-cells, the VISTA domain is found in resting T-cells, allowing researchers to prime the resting immune system to mount immune responses to cancer. Furthermore, VISTA is expressed in more cells in the TME than other members of the B7 family, giving it a larger cell pool to target.

CA-170, targeting both VISTA and PD-L1 in a phase 2 trial, showed a clinical benefit rate of 75% and progression-free survival of 19.5 weeks in patients with nonsquamous non-small cell lung cancer (NSCLC) [43]. Other current trials are mostly in the first phase and have been successful; however, VISTA’s expression in multiple cell lines has certain drawbacks, too. For example, VISTA expression is implicated in longer median OS in certain cancers such as hepatocellular and esophageal cancer when compared to VISTA-negative cell lines. Overall, this domain does have potential but still has some hurdles to overcome in regard to its applicability as a widespread checkpoint-based immunotherapy.

### 6.3. LAG-3

LAG-3 (lymphocyte activation gene 3) is a transmembrane protein expressed on various immune cells, including CD4+ and CD8+ T-cells, B-cells, and natural killer cells. LAG-3 has shown a significant role in regulating T-cell activation and proliferation by binding to MHC Class II molecules [41,44]. However, overexpression of LAG-3 can lead to T-cell exhaustion, allowing tumors to evade the immune system. Increased expression of LAG-3 has been found in ESCC tumor tissues compared to normal tissues [45,46].

Relatlimab is a human IgG4 monoclonal antibody that targets LAG-3. As of 2022, a combination of nivolumab plus relatlimab (Opdualag) has been approved for use in the U.S. for treatment of advanced metastatic melanoma in patients older than 12 years. In the RELATIVITY-047 trial, Opdualag monotherapy had an increased median PFS compared to just nivolumab (10.1 vs. 4.6 months; *p* = 0.006) in patients with advanced melanoma [47]. Although there are no definite conclusions of this therapy in esophageal cancer, a current clinical trial (NCT03044613) is studying the effects of neoadjuvant nivolumab plus relatlimab given concurrently with chemoradiation in patients with esophageal and gastric cancer.

Several trials are currently underway investigating the efficacy and safety of other LAG-3 inhibitors, such as eftilagimod alpha, fianlimab (REFN3767), and ieramilimab (IMP701) [48]. However, these studies are still in their early stages, and no conclusions can be drawn from them.

## 7. Conclusions

Esophageal cancer is a challenging disease with a high mortality rate. However, progress in immunotherapy has opened new avenues for treatment. Recent studies have shown drugs targeting PD-1 and CTLA-4, cell surface proteins that inhibit immune response, to be effective in treating locally advanced, unresectable, metastatic esophageal cancer. Many of these drugs significantly improved outcomes such as overall survival (OS) and overall response rate (ORR) in patients receiving immunotherapy compared to patients treated with only traditional chemotherapy. These treatments showed promise in neoadjuvant and adjuvant settings, as well as in conjunction with chemotherapy and other immunotherapy drugs.

Furthermore, emerging research has identified other biomarkers, including TIGIT, VISTA, and LAG-3, as potential targets in future immunotherapies, and some studies have already begun testing these drugs in treating cancers such as melanoma, yielding promising results. However, as is the case with PD-1 and CTLA-4 inhibitors, further studies must be conducted to fully illuminate the safety and efficacy of these therapies for patients with esophageal cancer. As we continue to make strides on this front, we will hopefully further improve the outcomes and quality of life for patients with advanced stages of esophageal cancer.

## Figures and Tables

**Figure 1 cancers-17-00851-f001:**
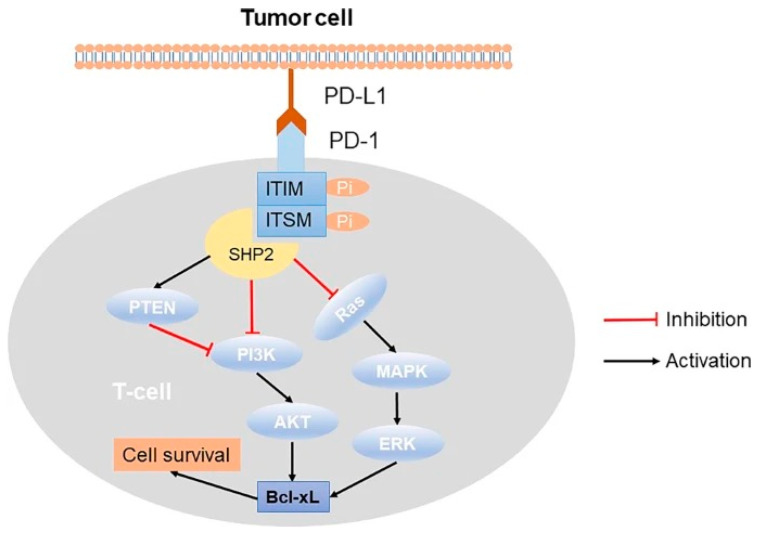
Immune pathway of PD/PD-L1 pathway. The binding of PD-1 and PD-L1 leads to the phosphorylation of the ITIM and ITSM motifs in the intracellular domain of PD-1. This phosphorylation recruits the tyrosine phosphatase SHP-2, which inhibits the PI3K/AKT and Ras/MAPK/ERK signaling pathways, resulting in T-cell death [10].

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
