# Peer review of "Current Advances in Immunotherapy Management of Esophageal Cancer"

_cancers, 2025, doi:10.3390/cancers17050851_

Round 1
Reviewer 1 Report
Comments and Suggestions for Authors
This is a review on esophageal cancer, which has one of the highest morbidity and mortality rates in the world. I think it is a particularly good summary of immunotherapy for esophageal cancer treatment. Each of the immunotherapy drugs, such as pembrolizumab, camrelizumab, cincilizumab, and tripalizumab, is briefly discussed. The book also provides a good summary of programmed T-cell therapies and prognostic biomarkers. In addition, new targets for immunotherapy such as Tigit and VISTA and LAG-3 are described in an easy-to-understand manner.
Author Response
Thank you for reviewing our manuscript titled “Current Advances in Immunotherapy Management of Esophageal Cancer”.
We are pleased that our manuscript is being reconsidered for submission. We thank the reviewers for their valuable feedback and encouraging comments. As requested, we have provided a point-by-point response to the comments below with relevant changes to the manuscript.
Comment: "This is a review on esophageal cancer, which has one of the highest morbidity and mortality rates in the world. I think it is a particularly good summary of immunotherapy for esophageal cancer treatment. Each of the immunotherapy drugs, such as pembrolizumab, camrelizumab, cincilizumab, and tripalizumab, is briefly discussed. The book also provides a good summary of programmed T-cell therapies and prognostic biomarkers. In addition, new targets for immunotherapy such as Tigit and VISTA and LAG-3 are described in an easy-to-understand manner."
Response: We appreciate the reviewer’s positive feedback.
Reviewer 2 Report
Comments and Suggestions for Authors
-
Limited Generalizability of Findings:
- A significant number of the clinical trials and studies referenced in the article were conducted exclusively in Chinese populations (e.g., trials involving sintilimab, toripalimab, and camrelizumab). This limits the generalizability of findings to broader, diverse populations and reduces the ability to draw conclusions applicable to global settings.
-
Over-reliance on Early-Stage or Preclinical Data:
- Many of the immunotherapies discussed (e.g., TIGIT, VISTA, LAG-3) are in early-stage trials or preclinical studies. While promising, the article sometimes overly emphasizes their potential without adequately addressing the limitations, such as small sample sizes and lack of robust phase III trial data.
-
Incomplete Discussion of Side Effects and Toxicity Management:
- While the article acknowledges treatment-related adverse events, it does not delve deeply into strategies for managing these toxicities. A detailed exploration of side effect mitigation could strengthen the discussion and provide practical insights for clinicians.
-
Lack of Cost-Effectiveness Analysis:
- Immunotherapies are often expensive. The article does not address the economic implications of these treatments, which are crucial for assessing their feasibility in resource-limited settings.
-
Insufficient Focus on Combination Therapies:
- Although the article mentions some combinations of immunotherapy and chemotherapy (e.g., pembrolizumab with chemotherapy), it does not systematically explore the potential of combining immunotherapies with other novel or conventional approaches, such as radiation therapy or other biological agents.
-
Minimal Discussion on Biomarker Limitations:
- While the article discusses biomarkers like PD-L1, ctDNA, and others as promising tools for guiding immunotherapy, it does not address the challenges of using these biomarkers in clinical practice, such as variability in testing methods and predictive accuracy.
-
Missed Opportunities for Comparative Analysis:
- The article presents data on various drugs like pembrolizumab, nivolumab, and toripalimab but does not perform a direct comparative analysis of their efficacy, safety, or cost. A systematic comparison could provide readers with a clearer understanding of the relative benefits and limitations of each option.
-
Lack of Future Perspectives or Emerging Trends:
- Although the article highlights some emerging biomarkers, it does not provide a forward-looking perspective on the evolving field of immunotherapy. Insights into cutting-edge research areas, such as personalized immunotherapy or AI-driven treatment planning, could add depth to the conclusion.
The English could be improved to more clearly express the research.
Author Response
Comment 1: A significant number of the clinical trials and studies referenced in the article were conducted exclusively in Chinese populations (e.g., trials involving sintilimab, toripalimab, and camrelizumab). This limits the generalizability of findings to broader, diverse populations and reduces the ability to draw conclusions applicable to global settings.
Response 1: Thank you for your comment. We have added a section (3.1.6) regarding tislelizumab's approval in Europe, North America, and Oceania.
Comment 2: Many of the immunotherapies discussed (e.g., TIGIT, VISTA, LAG-3) are in early-stage trials or preclinical studies. While promising, the article sometimes overly emphasizes their potential without adequately addressing the limitations, such as small sample sizes and lack of robust phase III trial data.
Response 2: We appreciate the reviewer’s comment. The necessary changes has been made throughout the manuscript to address the reviewer’s comment.
Comment 3: While the article acknowledges treatment-related adverse events, it does not delve deeply into strategies for managing these toxicities. A detailed exploration of side effect mitigation could strengthen the discussion and provide practical insights for clinicians.
Response 3: We appreciate the reviewer’s comment. In this article, we provided a comprehensive review of Immunotherapy medications being used in management of Advanced Esophageal cancer, their mechanism, application and potential side effects. The mitigation of side effects of the drugs is outside of the scope of this manuscript.
Comment 4: Immunotherapies are often expensive. The article does not address the economic implications of these treatments, which are crucial for assessing their feasibility in resource-limited settings.
Response 4: We appreciate the reviewer’s comment. A cost-based analysis is outside of the scope of this manuscript.
Comment 5: Although the article mentions some combinations of immunotherapy and chemotherapy (e.g., pembrolizumab with chemotherapy), it does not systematically explore the potential of combining immunotherapies with other novel or conventional approaches, such as radiation therapy or other biological agents.
Response 5: We thank the reviewer for the excellent comment. Additional information regarding combination therapy has been added to section 3.1.5 to address the reviewer’s comment.
Comment 6: While the article discusses biomarkers like PD-L1, ctDNA, and others as promising tools for guiding immunotherapy, it does not address the challenges of using these biomarkers in clinical practice, such as variability in testing methods and predictive accuracy.
Response 6: Thank you for your comment. Changes were made throughout the manuscript to address the reviewer’s comment
Comment 7: The article presents data on various drugs like pembrolizumab, nivolumab, and toripalimab but does not perform a direct comparative analysis of their efficacy, safety, or cost. A systematic comparison could provide readers with a clearer understanding of the relative benefits and limitations of each option.
Response 7: We appreciate the reviewer’s comment. A cost-based analysis is outside of the scope of this manuscript.
Comment 8: Although the article highlights some emerging biomarkers, it does not provide a forward-looking perspective on the evolving field of immunotherapy. Insights into cutting-edge research areas, such as personalized immunotherapy or AI-driven treatment planning, could add depth to the conclusion.
Response 8: Thank you for your comment. In response to personalized immunotherapy, Section 4 of the paper discusses CAR-T cell therapy along with current challenges with its administration. In regard to AI-driven treatment planning, we believe significant innovation in the field of immunotherapy with the use of AI is done through the use of NNs in receptor identification and target selection. However, this work is mostly done by labs focusing heavily on computational biology and is outside of the scope of this paper.
Comment 9: The English could be improved to more clearly express the research.
Response 9: Thank you for your comment. We have made changes throughout the manuscript to address the reviewer’s comment.